# Typology of Irrigation Technology Adopters in Oil Palm Production: A Categorical Principal Components and Fuzzy Logic Approach

**Diana Martínez-Arteaga** [1,2,*], **Nolver Atanasio Arias Arias** [1], **Aquiles E. Darghan** [2], **Carlos Rivera** [2] **and Jorge Alonso Beltran** [1]

1    Colombian Oil Palm Research Center-Cenipalma, Bogotá 11121, Colombia; narias@cenipalma.org (N.A.A.A.); jbeltran@cenipalma.org (J.A.B.)
2    Department of Agronomy, Faculty of Agricultural Sciences, Universidad Nacional de Colombia, Bogotá 11132, Colombia; aqedarghanco@unal.edu.co (A.E.D.); caariveramo@unal.edu.co (C.R.)
\*    Correspondence: dmartinezar@unal.edu.co; Tel.: +1-(786)5994082

**Abstract:** Oil palm is the second most cultivated oilseed crop in the world after soybeans, with more than 23 million hectares cultivated worldwide; it has become crucial for the economy of many countries. In Colombia, it is one of the most developed agricultural sectors, and every year the sector promotes the development of technologies that lead to greater sustainability of agricultural and food systems and address the challenges and opportunities of agribusiness. In this research, the central focus was the adoption of irrigation technologies, which is limited despite significant efforts and investments in physical and human capital. On many occasions, the typology of farmers has been associated with low technology implementation. Thus, linking the typology of farmers according to certain commonalities or differences is an essential step in exploring the factors that explain the adoption. In addition, the ranking also helps in the understanding of existing adoption constraints, as well as finding opportunities for change. This study aimed to determine the socioeconomic and demographic typology of those who adopt irrigation technologies. The analysis was performed using categorical principal component analysis to reduce dimensionality and fuzzy cluster analysis to classify the groups. As a result, four groups of producers that differ in terms of their demographic and socioeconomic characteristics were obtained, where the groups "population with female leadership" and "diversified population" were the adopters of irrigation technologies. The most outstanding characteristics of these two groups were the profitability of the harvest and the age of the producers. Determining the typology of farmers is a fundamental step in expanding the technology adoption process through agricultural extension services, which represent a way of reaching producers directly. In addition, these results allow decision makers to participate in this dynamic reflectively and intentionally (such as governments, researchers, and technology transferors).

**Keywords:** farming system; sustainability agricultural; smallholder; technology adoption; oil palm

## 1. Introduction

Palm oil has become an essential commodity, and its growing demand is associated with its requirement in the food, cosmetics, soap, pharmaceutical, and biofuel industries [1]. Consequently, the demand for 2050 is forecast to be 240 million tons per year [2]. This agribusiness has driven rapid economic growth in several tropical countries, making it an essential contributor to alleviating rural poverty and food and energy independence [3]. The World Bank Group has a mission to reduce poverty and believes that this commodity can be essential in promoting economic development in these countries and improving the standard of living of the rural poor. Thus, the oil palm agroindustry addresses social, economic, and governance risks and contributes to global food security. Consequently, this product benefits the livelihoods of many communities, the GDP of countries, and

the achievement of several sustainable development goals (SDGs), including eliminating poverty, hunger, and promoting decent work and economic growth [4].

Palm oil is derived from oil palm cultivation, which is performed in developing countries in the humid tropics, often forming an important base for local economies, both as an export and as a raw material for local industry. The area used for oil palm cultivation increased from less than 5 million hectares in 1980 to more than 20 million hectares in 2018, with 50% of the cultivated area worldwide being managed by small producers [5]. In some countries, small farmers use 94% of their land to cultivate oil palm [6].

Colombia is the largest producer of palm oil in Latin America and the fourth largest producer worldwide, contributing 2.2% (1.7 MnT) of the world's palm oil compared with Malaysia and Indonesia, which together produce 82.3% (64.9 MnT) (Oil World, https://www.oilworld.biz/, accessed on 6 June 2023) [7]. The growing presence of Colombian palm oil in international markets makes it imperative to differentiate the environmental, labor, and economic practices that have stigmatized oil palm cultivation. The country has close to 30% of its production certified with one of the international sustainability seals, which makes it a leader on this front at a global level [8]. In a global study that statistically estimated the growth of palm between 1989 and 2013 in producing countries and regions and its contribution to deforestation, a rate of zero (0) net deforestation was identified in Colombia despite a palm growth of 69.5% during this period in which there was high expansion [9].

With 600,000 planted hectares, oil palm is currently the second most important crop in terms of extension in Colombia and has around 6856 producers, 85% of whom are small-scale and 12% medium. This is an agroindustry that has developed around farmers and formal companies, generating quality employment opportunities at the local level, and bringing invaluable social and economic development to much of the country [8]. In the social approach, currently, the fruit suppliers are mostly small producers that cultivate 70% of the planted area. The growing participation of small and medium palm growers has contributed to alleviating poverty and reducing the levels of rural violence associated with conflicts over land and the production of illicit crops [10].

However, the participation of small and medium palm growers, in turn, has translated into new challenges such as low crop yields, low technology adoption, and rural extension, among others. Colombia produces a slightly higher yield (3.50 t oil/ha) than Indonesia (3.30 t oil/ha) and Malaysia (3.49 t oil/ha); these figures are far from the sectoral goals of 23 tons of fruit and 5 tons of oil per hectare of the average country [8]. The explanation for this gap has to do with multiple significant factors, including the water deficit in some areas (especially the north of the country) due to conditions related to climate change and variability.

There is increasing evidence to suggest that climate change is real and has potentially devastating consequences for humanity [11]. Consequently, environmental variability due to climate change challenges the adaptive capacity of socio-ecological communities in regions dependent on climate-sensitive agriculture [12,13]. The application of efficient irrigation (drip and sprinkler) methods can help alleviate the effects of climate change and protect water resources, especially in areas with water scarcity [14], allowing a reduction in the impacts associated with climate change and variability [15,16]. However, in the case of the north zone (Study area), about 90% of the area planted with oil palm has low-efficiency irrigation application methods, which seek to address the water deficit, but do not complement the low environmental supply [17]. In addition, in the long term, these inefficient systems are unsustainable due to their high water consumption and the low availability of water resources, which implies that producers must consider the efficiency of the water use system when irrigating their crops [18,19].

The adoption of technologies is particularly relevant [20]. However, the adoption of technologies is the result of a complex, dynamic, and interactive process that occurs within a heterogeneous set of actors [13]. In addition, there is a mismatch between the available technologies and the socio-economic circumstances of the farmers since the adoption

decisions depend on the diverse needs and abilities of the farmers [11]. Within literature, there is a significant body of work that looks at the critical drivers of the adoption of agricultural technology and technology diffusion. One of these drivers is the characteristics or typologies of the producers (for example, education, age, and income) [10]. Consequently, the typology of farmers is associated with the implementation of technologies and crop yields [21].

Studies on influential factors in the adoption decision have been conducted, and the results or variables associated with the adoption of technologies vary from one study to another. The central hypothesis is that the adoption of innovations depends significantly on the characteristics of the producer. Some of these studies include the following: "Influence of the profile of producers in the adoption of innovations in three tropical crops are highlighted" [22]; "Information networks that generate economic value: a study on groups of adopters of new or improved technologies and practices among oil palm producers in Mexico" [23]; "Do wealthy farmers implement better farming practices? An evaluation of the implementation of Good Agricultural Practices among different types of independent smallholder oil palm farmers in Riau, Indonesia" [21]; and "A typology of adopters and non-adopters of improved sorghum seeds in Tanzania: a deep learning neural network approach" [24] among others. The results are a guide for extension workers on the most predominant factors. However, the results should not be generalized, but used as a reference based on the characteristics of the farmer and their local and technological specificities. This study aimed to determine the socioeconomic and demographic typology of irrigation technology adopters located in a region with environmental risks.

The results of this study expand the available conceptual framework and contribute to a better understanding of the reality of oil palm producers in Colombia. Under the references, the typology of producers helps to understand the existing adoption limitations, as well as to find opportunities for change. Among these changes, extension services can help guide type-specific management considering the distinction between the formed groups. In addition, it allows the development of projects and actions to improve the response conditions to the challenges faced by oil palm producers (government, researchers, technology transferors).

The document is organized as follows. Section 2 presents the description and the research methods used; Section 3 presents the results that arose from the statistical analysis. The corresponding discussion is presented in Section 4, and finally, the conclusions are in Section 5.

## 2. Materials and Methods

This study was observational and field surveys via face-to-face interviews with 131 producers (area of 3.200 ha) located in the Sevilla River basin, in the department of Magdalena, were used (Figure 1). The questions were oriented toward the structuring of the basic social indicators, that is, conditions of palm growers and sociodemographic indicators. In addition, farmers' responses to the irrigation systems used in their plantations were considered. In detail, each theme did or did not incorporate irrigation technologies over time. The data source was a survey carried out by the Oil Palm Research Center (Cenipalma) in May and July 2021. The information was recorded in a mobile application designed by Cenipalma.

To treat the generated data matrix, a categorical principal components (CPC) analysis was used (Gifi 1990) [25], which is a particular form of nonlinear principal components analysis (PCA) based on the categorical coding of the variables in indicator matrices $G_1, \ldots, G_m$ that satisfies the following proportionalities:

$$x \propto m^{-1} \sum_j G_j y_j$$

$$y_j \propto D_j^{-1} G_j' x$$

where x is the object scores vector and $y_j$ is the quantification vector of the categories of the j-th variable. D is defined as the partitioned diagonal matrix of C and C is called "tableau de Burt". G is called the indicator matrix of $h_j$, where $h_j$ corresponds to the j-th column of the H matrix, and the data matrix H is an n × m matrix. The steps implemented in the R software, version 4.1.3, involved updating the score vector, normalization, updating the quantification of the categories, and the convergence test.

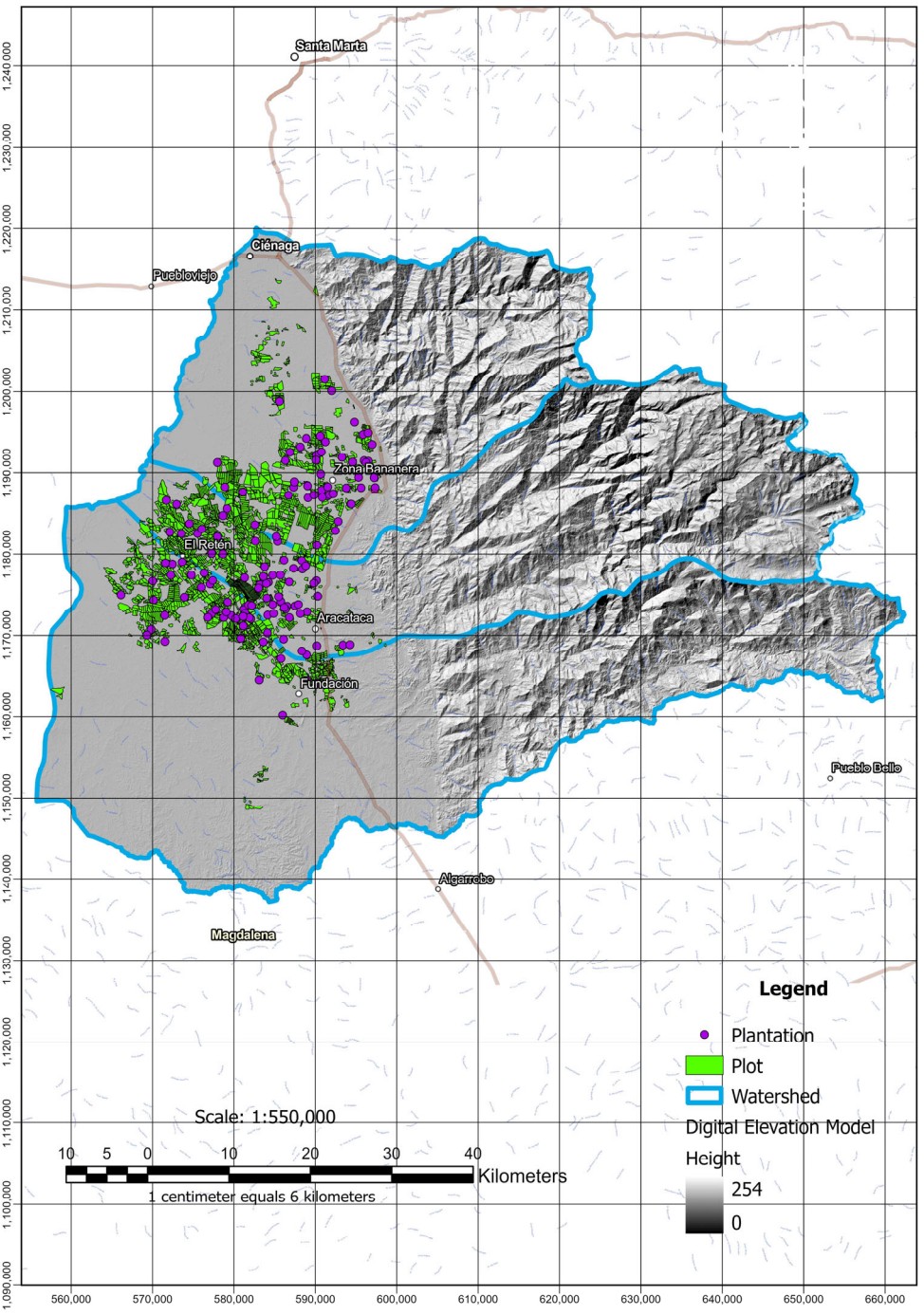

**Figure 1.** Geographical location of the study area.

This method allows the extraction of the contributions by variables or observations in a manner analogous to the usual PCA. With these contributions and the variance explained by the extracted dimensions, metric information of the contributions per row was available. The number of dimensions that reached a certain threshold formed the new data matrix.

It is important to mention that currently in the social and behavioral sciences, these multivariate techniques provide interesting visualizations for complex data obtained from surveys [26]. Each question corresponds to a variable, and each possible answer is a category of that variable [27]. For the CPC, 15 variables were selected corresponding to the size of the farm, land use, type of land tenure, crop profits, type of workers hired per year, economic activities, age, the producer's gender, education, adoption of irrigation technologies, and, finally, if agricultural practices such as the application of irrigation, fertilizers, and pest and disease management schemes are used (Table 1).

**Table 1.** Variables surveyed describe the characteristics of the producers and the plantations.

| Category | Variable | Unit |
|---|---|---|
| Demography and socioeconomics | Age | 1 = <30, 2 = 31–50, 3 = 51–65, 4 = 66–80, 5 = >81 |
| | Sex | 1 = Male, 2 = Female |
| | Can read and write | No, yes |
| | Level of education | 0 = None, 1 = Low (primary), 2 = Medium (secondary) |
| | Higher Education Degree | 0 = None, 1 = Technician or Technologist, 2 = Professional |
| Farm characteristics | Farm size—Fedepalma category | 1 = <50 ha, 2 = 51–500 ha |
| | Land use | 1 = Exclusively for production, 2 = Production and permanent residence |
| | Type of holding | 1 = Inheritance succession, 2 = Owner with title, 3 = Holder, 4 = Tenant, 5 = Co-owner with title |
| Productive activities | Works in activities other than cultivation | No, yes |
| | Type of workers | 1 = Relatives–Employees, 2 = Relatives, 3 = Employees |
| | Profits from palm cultivation | 1 = From COP 0 to COP 500.000, 2 = From COP 500.000 to COP 1500.000, 3 = From COP 1500.000 to COP 3000.000, 4 = From COP 3000.000 to COP 6000.000, 5 = More than COP 6000.000 |
| Cultivation characteristics | Uses irrigation system | No, yes |
| | Adoption of irrigation technologies | 1 = If the farmer has surface/flood irrigation, 2 = If the farmer adopts efficient irrigation technologies (drip or sprinkler) |
| | Uses fertilizers | No, yes |
| | Has a pest and disease management scheme | No, yes |

Then, fuzzy cluster analysis (FCA) was used to form groups with individual contributions. The FCA algorithm attempts to partition a finite collection of $n$ elements $X = \{x_1, \ldots, x_n\}$ into a collection of c (cluster centers $C = \{c_1, \ldots, c_n\}$) fuzzy clusters with respect to some given criterion and a partition matrix $W = w_{ij} \in [0, 1]$, $i = 1, \ldots, n$, $j = 1, \ldots, c$, where each element, $w_{ij}$, communicates the degree to which the element, $x_i$, belongs to the cluster $c_j$. The idea is to minimize the function.

$$J(W, C) = \sum_{i=1}^{n} \sum_{j=1}^{c} w_{ij}^m \left\| x_i - c_j \right\|^2$$

where m is the hyper-parameter that controls how fuzzy the cluster will be. Finally, $w_{ij}$ is defined by

$$w_{ij} = \frac{1}{\sum_{k=1}^{c} \left( \frac{\left\| x_i - c_j \right\|}{\left\| x_i - c_k \right\|} \right)^{\frac{2}{m-1}}}$$

where $\|.\|$ is the Euclidean norm.

FCA is more efficient in some investigations, and the clustering results are better than other methods. However, FCA has a disadvantage in that clustering results are affected by

clustering configurations, such as the number of clusters [28]. Consequently, the algorithm requires a "fuzziness" or result validation parameter. An algorithm was developed to optimize this parameter [29] and to select the number of groups to build. The Xie Beni (XB) index method introduced by Xie and Beni is one of the most popular group validation methods [30,31]. The XB focuses on the proximity of the data in one group and the distance between the center of one group and the other. The smallest XB value shows the best number of clusters. Once the parameter was chosen by perfecting the index, the cluster analysis method was applied to end the groups' typing or characterization.

Cluster analysis is an exploratory tool designed to discover the natural groupings, thus allowing the generation of information criteria and identification of homogeneous groups (clusters) of objects and/or individuals in a database [32]. Consequently, when the intra-cluster similarity is more significant, the nature of the grouping can be better understood and studied.

The univariate descriptive analysis table was created only in an exploratory way since the analysis was finally carried out in a multivariate way Table 2 shows the percentage distribution by categories previously described in Table 1 of each of the variables.

**Table 2.** Univariate descriptive analysis by categories of each of the variables.

| Variable | Response | | | | | | | |
| --- | --- | --- | --- | --- | --- | --- | --- | --- |
| | 0 | 1 | 2 | 3 | 4 | 5 | NO | YES |
| Age | | 1% | 29% | 34% | 27% | 8% | | |
| Sex | | 78% | 22% | | | | | |
| Can read and write | 5% | 95% | | | | | | |
| Level of education | 5% | 51% | 44% | | | | | |
| Higher Education Degree | 56% | 16% | 28% | | | | | |
| Farm size- Fedepalma category | | 96% | 4% | | | | | |
| Land use | | 56% | 44% | | | | | |
| Type of holding | | 7% | 53% | 38% | 1% | 1% | | |
| Works in activities other than cultivation | | | | | | | 72% | 28% |
| Type of workers | | 64% | 28% | 8% | | | | |
| Profits from palm cultivation | | 11% | 52% | 24% | 8% | 5% | | |
| Uses irrigation system | | | | | | | 2% | 98% |
| Adoption of irrigation technologies | | 85% | 15% | | | | | |
| Uses fertilizers | | | | | | | 7% | 93% |
| Has a pest and disease management scheme | | | | | | | 11% | 89% |

## 3. Results

The analysis of categorical principal components showed that six components accounted for 70.8% of the data variability, from which the matrix of scores of the components was extracted (Supplementary Materials) (Figure 2).

For the optimization of the cluster number and fuzziness parameters, the Xie and Beni indices were compared in different scenarios, extracting the one with the smallest index. With the index matrix, the minimum was found in cluster 4. Therefore, four clusters were identified using the group validation method introduced by Xie and Beni (XB). These groups were the result of the proximity of the data in one group and the distance between the center of one group and the other [28].

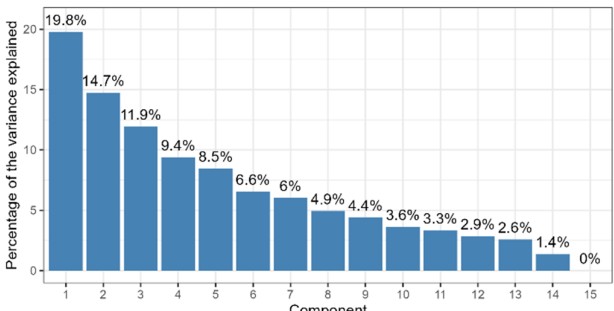

**Figure 2.** Variance percentages expressed by dimensions resulting from the revealed data that describe the characteristics of the producers and plantations.

The first group—G1 (Vulnerable population) was formed by 43 producers, of which 2 were adopters of irrigation technologies. The second group—G2 (Population with female leadership) had 29, with 9 adopters, the third—G3 (Diversified population) 23 producers with 6 adopters, and the fourth—G4 (Population with exclusivity in oil palm crops) 36 producers and only 3 adopters. Next, the distinctive features of the groups are described individually and are summarized in Figure 3.

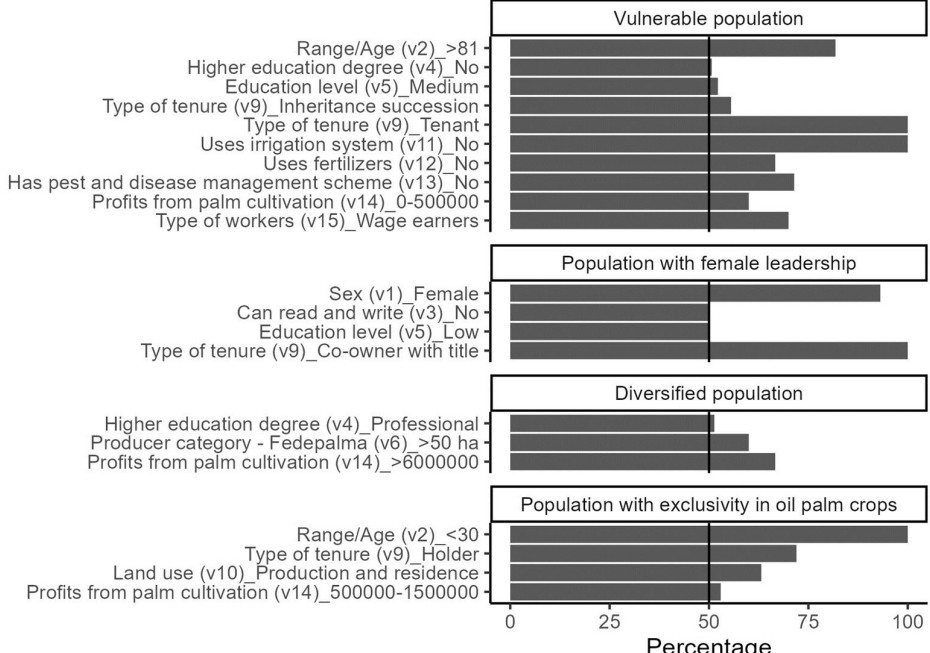

**Figure 3.** Percentages of the variables by groups of oil palm producers in the North of Colombia.

Vulnerable population: Group 1, with 33% of the producers in the sample, comprised only small-scale farmers (<50 ha) according to the definition of a small owner (Rspo.org, accessed on 6 June 2023) [33]. They were all men aged between 31 and 94 years and included producers with older generations (81 to 94 years, 7%). Likewise, 32.23% reported low educational levels, corresponding to primary or secondary education. In terms of their production units, the use of the plantations is primarily to produce oil palm. More than half were dedicated exclusively to cultivation activities; a large part were owners with titles to their properties. The main difference in this group is that its members were all men; however, it was the most heterogeneous group in terms of employment. In this sense, the sources of labor were distributed almost equally between family members and external sources. In addition, the profits from the crop ranged from COP 500,000 to more than COP 3,000,000 per month. Regarding the use of agricultural practices such as the application of irrigation, fertilizers, and pest and disease management schemes, most of the producers

applied these to their crops; however, despite using irrigation in their plantations, 31.46% did so inefficiently with traditional methods (surface/flood irrigation).

Population with female leadership: Group 2 comprised 22% of the producers. This group was characterized by several women leading agricultural production (20.48%) and comprised only small-scale farmers (<50 ha). The average age of these producers was 61 years. Additionally, producers who did not know how to read and write were found. However, the majority reported secondary education levels (15.17%). As for their productive units, more than half had their plantations producing palm oil, and the income from this agroindustry comprised 15.93%. Most of them were owners with titles to their properties and which employed both family and salaried external labor. In addition, the revenue generated by oil palm ranged from COP 500,000 to COP 6,000,000 per month. Finally, 22% of the G2 producers applied irrigation in their plantations, among which 6.82% adopted efficient irrigation systems (drip or sprinklers). Of the 6.82% adopters, 5.31% were women. Regarding fertilization practices and pest and disease management schemes, 20.48% implemented them.

Diversified population: Group 3 comprised 17.55% of the sample; they were all men. The producers of this group were the youngest, with an average age of 57 years. Regarding the educational level, 14.49% of the producers were professionals. Consequently, they were the producers with the greatest abilities to decode explicit knowledge according to the levels of schooling. Regarding the characteristics of the farms, the vast majority were producers with plantations of less than 50 hectares dedicated exclusively to the production of oil palm. Consequently, this agroindustry was the source of their entire income; they were owners with formal titles to their properties. Regarding the type of workers, more than half of the workforce employed in these farms came from external and family sources. The profits received by these producers were between COP 1,500,000 and COP 6,000,000 per month per ton of palm. Regarding the practices of irrigation, fertilization and pest control, and disease management, 16.78% implemented them. However, only 4.57% were adopters of efficient technologies. However, this was one of the groups with the highest adoption of irrigation technologies.

Population with exclusivity in oil palm crops: Group 4 comprised 27.48% of the producers. Most were men between the ages of 31 and 81. In addition, they were the ones with the highest education levels. The main difference between this group was that many variables had the highest values in this group, such as the use of plantations (agricultural production and permanent residence), people working only in oil palm cultivation, owners with titles to their properties, and the sources of work being distributed between families and external sources, and the profits from the crops were between COP 500,000 and COP 1,500,000 per month. In addition, all of them applied irrigation, fertilization and pest control, and disease management schemes. However, only 2.29% of the producers adopted efficient technologies for managing water resources.

## 4. Discussion

The characterization of the common typologies of oil palm producers located in Northern Colombia and the description of aspects that cause differentiation are conducted to support for technology adoption through extension support systems for each group. From the results of this study, a greater adoption of technologies was observed in the groups that presented a better generation of economic value, that is, more profits from the crop. In addition, as age increases, the adoption of technologies decreases, that is, producers of older ages have a lower propensity to innovate and young producers are more receptive to innovation [13,22].

In this sense, it was found that the adoption of technologies depends on several variables that agree with previous works on technological adoption [23]. The most highlighted by their frequency distribution or associated statistics were the age of the producers, the income obtained from production, the size of the plantation, and the use of their plantations. Regarding this last variable, greater adoption was observed when the plantations only had

productive purposes. It is worth mentioning that the use of pressurized irrigation allows farmers to increase the efficiency of water use, increase the quantity and quality of crops, and reduce costs. Consequently, producers who are farther removed from their plantations invest in technologies that increase time consumption, labor intensity, and the waste of water resources.

A high implementation of irrigation, fertilization, and pest and disease management schemes was found among different groups. However, low adoption of water efficiency technologies was found. It is worth mentioning that adoption was measured by the degree of use of the new technology in long-term, that is, the scope and intensity of the use of technology in place of the initial decision to adopt a new practice in a short time [13,34].

The results revealed a homogeneous distribution in variables such as the type of tenure, activities exclusive to palm cultivation, type of workers, and number of hectares. Land tenure is important, considering that oil palm is a perennial crop (with approximately 30 years of production cycles) and requires long-term investments. The sources of work were shared almost equally between family and external sources. However, the social initiatives of labor formalization are a factor to consider, considering the productive dynamics and the growing activity of the crop in Colombia.

Due to the knowledge of the heterogeneity of the producers and their characteristics, it is more effective to research and extend the training of farmers in the adoption of water efficiency technologies. In addition, the identified groups make it possible to broaden the adoption process through targeted extension services and those specific to management groups. Likewise, researchers should be involved in developing pressurized irrigations to alleviate some of the obstacles to their adoption.

## 5. Conclusions

As a result, four groups of producers were obtained that differ in terms of their demographic and socioeconomic characteristics, where the groups "population with female leadership" and "diversified population" were the most frequent adopters of irrigation technologies. In addition, the most outstanding characteristics of these two groups are the profitability of the harvest and the age of the producers.

It is important to consider the group of "vulnerable producers". In particular, the socioeconomic characteristics of these users may indicate the potential ability to access resources, services, and differentiated technical support.

Based on the results, technology transfer strategies should consider the typology of producers and the groups formed to expand the adoption process through extension services, that is, develop type-specific management schemes considering the distinction between them.

In the past, many research studies on influential factors on the decision to adopt technologies have been conducted; the results and variables associated with the adoption of technologies vary from one study to another. Therefore, the results should not be generalized, but used as a reference based on the characteristics of the farmer, the locality, the technologies, and the agricultural sector.

The adoption of technology is a complex process, and the variables are not limited to those studied; for future research, it is recommended to include, as part of the model, the categories of technical assistance, access to credit, and experience, among others, to strengthen the analysis of this classification and to establish causal relationships.

**Supplementary Materials:** The datasets generated during and/or analyzed during the current study are available in the PRINCALS-FuzzyCluster-MartinezDarghanRivera repository, https://github.com/CarlosRivera1212/PRINCALS-FuzzyCluster-MartinezDarghanRivera, accessed on 6 June 2023.

**Author Contributions:** Conceptualization, D.M.-A.; data curation, D.M.-A.; formal analysis, D.M.-A., A.E.D. and C.R.; funding acquisition, N.A.A.A. and J.A.B.; investigation, D.M.-A.; methodology, D.M.-A., N.A.A.A., A.E.D. and J.A.B.; project administration, N.A.A.A., J.A.B. and D.M.-A.; software, D.M.-A., A.E.D. and C.R.; supervision, N.A.A.A. and J.A.B.; validation, A.E.D. and C.R.; Visualization,

D.M.-A., A.E.D. and C.R.; writing—original draft, D.M.-A.; writing—review and editing, D.M.-A., N.A.A.A., A.E.D., C.R. and J.A.B. All authors have read and agreed to the published version of the manuscript.

**Funding:** This research was funded by the Colombian Oil Palm Promotion Fund (FFP) administered by Fedepalma (IPR0222).

**Institutional Review Board Statement:** The form kept the Personal Data Treatment Policy, which is published on the official websites of Fedepalma and Cenipalma (article 4 and 11 of Decree 1377 of 2013).

**Informed Consent Statement:** Informed consent was obtained from all subjects involved in the study.

**Data Availability Statement:** We made all data from the Characterization of Palm Growers conducted by CENIPALMA available at https://geoportal.cenipalma.org/wa/home/ accessed on 6 June 2023. All the information about Cenipalma investigations can be found at https://www.cenipalma.org/.

**Acknowledgments:** This work was supported by the Colombian Oil Palm Research Center-Cenipalma, especially in the extension and geomatics areas. Special thanks to the researchers Osmar Ricardo Barrera Agudelo, Wilmer Velasco Silva, and Oscar Sanabria Fernandez. Support was also received from the Colombian Oil Palm Promotion Fund (FFP) administered by Fedepalma.

**Conflicts of Interest:** The authors declare no conflict of interest.

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
