# Peer review of "Typology of Irrigation Technology Adopters in Oil Palm Production: A Categorical Principal Components and Fuzzy Logic Approach"

_sustainability, doi:10.3390/su15139944_

Round 1

Reviewer 1 Report

The purpose of the article was to understand the socioeconomic and demographic typology of the palm oil producers with a better adoption of irrigation technologies through categorical principle components analysis and fuzzy cluster analysis.

 The issue chosen is significant because it shows how effective irrigation may be used to adapt to climate change while also protecting water resources, particularly in dry regions. The process surrounding this problem is complicated, dynamic, and participatory.

My focus, though, has been on the key elements. Not only because of the writing's politeness, but also because of the following:

-          The text has a few grammatical and grammatical errors. It need a thorough rewrite.

-          The abstract needs to be updated to include more study findings.

Introduction:

-          Detailed specific objectives should be included at the end of the introduction.

-          Figure 1 needs to be replaced with a new one that explains the research roadmap because it is not clear.

Materials and method:

-          I suggest including a map that shows the farms used in the study.

-          Please provide specific references from the Food and Agriculture Organization (FAO), National Administrative Department of Statistics (DANE), and Economic Commission for Latin America and the Caribbean (CEPAL) that were used to gather the endorsed, socioeconomic and demographic indicators.

-          Add (CPC) after categorical principal components on page 3 line 133.

-          Please include the reference of (Gifi 1990).

-          Please complete the definitions of all equation symbols, including m, D, and subscript y, on page 4 line 140.

-          Please include the cluster analysis distance and Euclidean equations, as well as the cluster analysis chart, on page 4 line 168.

Results and discussion

Please provide the outcomes of the variables' descriptive statistical analysis.

Please provide PCA findings and analysis, including:

-          - Table of Pearson correlation matrix of different variables.

-          Table of rotated component matrix with factor loadings

-          Biplot vectors for PC1 and PC2

Figure 4's radar/spider plots require further explanation.

Conclusions

- More study findings should be included in the conclusions.

Minor editing of English language required

Author Response

Thank you very much for your valuable comments. The article was adjusted and improved.

Diana 

Reviewer 2 Report

Sustainability – Paper Review

Socioeconomic and Demographic Typology related to a Better 2 Adoption of Irrigation Technologies in the Palm Oil Sector in Colombia.

Paper Section

Comment(s)

Title

Please format the title properly. The first letter of connecting words or punctuations must be small.

Abstract

The Abstract of manuscript is very brief and self-explanatory to the readers. It clearly reflects the study design and brief about methodology used.

Just add some lines about basic purpose of the study in the start of abstract. It’s good that you have written a conclusive statement at the end of the abstract.

Introduction

The introduction section is very briefed and informative. It shows the commitment and devotion to work from researchers. Every paragraph is very elaborated very well and connected symmetrical. References are from latest studies. Two suggestions are suggested that references should be cross referenced and images size should be increased for reviewers ease.

Material and Methods

Material and methods are self-explanatory. Hard work done by researches showed in methodology. One correction is suggested, increase the size of equation mentioned in methodology.

Results

The results are clear and demonstrated well. Size of image should be increased. Text present in image is not cleared.

Discussion

The discussion done by researchers is excellent. Only the size of image should be increased.

Conclusion

The conclusion of study is showing the reflect of whole manuscript with research gape given in very well way.

General Comment

Overall, the manuscript seems excellent with brief methodology, results and discussion.

The citation and references styles are latest which is good.

Overall manuscript is extra-ordinary and recommended for publication after incorporating the comments and thorough proof reading. Manuscript needs thorough proof reading and corrections before publication.

Author Response

(The authors gave the same response as above.)

Reviewer 3 Report

Sustainability-2421573

I am pleased to submit my review for the manuscript titled: Socioeconomic and Demographic Typology related to a Better Adoption of Irrigation Technologies in the Palm Oil Sector in Colombia. I liked the research idea and its strength to the scientific community, especially those belong to palm oil industry. Although, please address these comments and recommendations carefully.

1.       In order to establish the significance and value of the study, it is necessary to provide a comprehensive rationale for the research, which emphasizes its relevance and unique contributions to the current scholarly discourse. This will strengthen the study's originality and academic impact.

2.       Please remove the sign (.) from the title.

3.       I suggest authors revise the title and make it attractive for the readers.

4.       The author must revise the abstract to include such gaps, how they fill them, topic importance, methods and clear-cut findings, and policy implications from research findings.

5.       The introduction is generally acceptable for the first part, but it fails to establish the need for this study. What was the driving force behind conducting this particular study?

6.       At the end of the introduction, in presenting the paper goals, try to answer the questions: How is the current research necessary? And how is it novel and contributes to the state of the art?

7.       The research gap should be clearly defined.

8.       The findings of the study need to be more elaborate. This section needs to be developed and supported by previous work. The discussion needs to be improvised with a theoretical contribution. The findings of the discussion need to be strengthened with the previous research work.

9.       The discussion needs to be improved with a theoretical contribution.

10.    The conclusion is weak. It should also be an extrapolation of the key findings from the research and not a summary. So, there should be conclusions around the background theory, data theory/analysis, and key outcomes. The authors should have included the following sub-sections within the conclusion section with more details:
*       Implications to theory and practice should be clearly stated;
*       Key lessons learned;
*       Limitations of this research;

11.    Future research directions should be improved; in that, they should stem from the awareness of the limitations and opening avenues related to the obtained outcomes.

12.    Proofread the whole manuscript as many typos and grammar errors are present.

Minor English editing is required. 

Author Response

(The authors gave the same response as above.)

Reviewer 4 Report

This study addressed socioeconomic and demographic typology related to adoption of irrigation technologies, the topic is reasonable but the expression of material and method and results sections needs to be improved and the discussion section requires extensive review and major revision as commented:

Abstract

Line 12: delete ‘and’

Line 14: ‘for’ change into ‘is’

Line 14-15: ‘In Colombia it is one of the most developed agricultural sectors and promotes technologies…’

Line 16-17: readdress the focus of this study.

Line 21-25: the results were not clearly defined, e.g. what are the four groups of producers?

Introduction

Line 64 and 73: does ‘7’ represent reference 7?

Line 71: does yield mean oil yield?

Line 81-82: ‘efficient irrigation (drip and sprinklers) method’

Line 85-86: not clear. Did author address that water deficit irrigation is the major irrigation adoption in study area and it decreases the yield?

Materials and Methods

In this section, the description of two analysis methods (CPC and FCA) didn’t seem very useful for readers to understand what were exactly analyzed using these two methods in this paper.

Line 140: ‘yj’, please put ‘j’ as subscript.

Table 1: the variables surveyed in this study were issued by internationally accredited institutions e.g. CEPAL, DANE, FAO, are these variables the sum of all the institutions or the author selected some of variables from each institution? If so, what is the selection criteria?

Results and Discussion

Needs to do order for paragraphs to improve results and discussion sections. Please move Figure 4 as well as description text to results section and add sub-titles in results.

Line 270: what does ‘said adoption’ mean?

Discussion needs to be expanded and compacted.

Moderate editing of English language required

Author Response

(The authors gave the same response as above.)

Round 2

Reviewer 1 Report

-          The most of grammatical errors have been corrected, but still some remean.

-          The abstract has been updated.

-          Figure 1 have been modified with a new one.

-          I suggested map that shows the farms used in the study, has been added.

-          The (CPC), the reference of (Gifi 1990), the definitions of all equation symbols, the cluster analysis distance, euclidean equations and univariate descriptive analysis were added.

- Result and conclusion were modified.

-          The most of grammatical errors have been corrected, but still some remean.

Reviewer 3 Report

Authors have incorporated all my concerns and recommendations.  The manuscript can be accepted for publication. 

Minor English correction is required. 

Reviewer 4 Report

Accept in present form.